# Transformed Waldenström Macroglobulinemia: Update on Diagnosis, Prognosis and Treatment

Eric Durot [1,*], Cécile Tomowiak [2], Elise Toussaint [3], Pierre Morel [4], Dipti Talaulikar [5], Prashant Kapoor [6], Jorge J. Castillo [7] and Alain Delmer [1]

1. Department of Hematology, University Hospital of Reims and UFR Médecine, 51092 Reims, France
2. Department of Hematology and CIC U1402, University Hospital of Poitiers, 86000 Poitiers, France
3. Department of Hematology, University Hospital of Strasbourg, 67200 Strasbourg, France
4. Department of Hematology, University Hospital of Amiens, 80480 Amiens, France
5. Department of Hematology, Canberra Health Services, College of Health and Medicine, Australian National University, Canberra, ACT 2600, Australia
6. Division of Hematology, Department of Internal Medicine, Mayo Clinic, Rochester, MN 55901, USA
7. Bing Center for Waldenström Macroglobulinemia, Dana-Farber Cancer Institute, Harvard Medical School, Boston, MA 02459, USA
* Correspondence: edurot@chu-reims.fr

**Abstract:** Histological transformation (HT) to an aggressive lymphoma results from a rare evolution of Waldenström macroglobulinemia (WM). A higher incidence of transformation events has been reported in *MYD88* wild-type WM patients. HT in WM can be histologically heterogeneous, although the diffuse large B-cell lymphoma of activated B-cell subtype is the predominant pathologic entity. The pathophysiology of HT is largely unknown. The clinical suspicion of HT is based on physical deterioration and the rapid enlargement of the lymph nodes in WM patients. Most transformed WM patients present with elevated serum lactate dehydrogenase (LDH) and extranodal disease. A histologic confirmation regarding the transformation to a higher-grade lymphoma is mandatory for the diagnosis of HT, and the choice of the biopsy site may be dictated by the findings of the [18]fluorodeoxyglucose-positron emission tomography/computed tomography. The prognosis of HT in WM is unfavorable, with a significantly inferior outcome compared to WM patients without HT. A validated prognostic score based on 3 adverse risk factors (elevated LDH, platelet count $< 100 \times 10^9$/L and any previous treatment for WM) stratifies patients into 3 risk groups. The most common initial treatment used is a chemo-immunotherapy (CIT), such as R-CHOP (rituximab, cyclophosphamide, doxorubicin, vincristine, prednisone). The response duration is short and central nervous system relapses are frequent. Whether autologous stem cell transplantation could benefit fit patients responding to CIT remains to be studied.

**Keywords:** diffuse large B-cell lymphoma; aggressive lymphoma; histological transformation; IgM lymphoplasmacytic lymphoma; $MYD88^{L265P}$ mutation

## 1. Introduction

The transformation of Waldenström macroglobulinemia (WM), an IgM lymphoplasmacytic indolent lymphoma, into an aggressive lymphoma was first described by Wood and Frenkel, who reported the development of multiple "lymphoblastic lymphosarcomatous" masses in a patient with WM in 1967 [1]. The first reports described histological transformation (HT) in WM as "reticulum cell sarcoma" or "immunoblastic sarcoma" [2–6]. In the study by Garcia and colleagues [7], two cases of transformed WM were analyzed along with 14 cases reported in the literature. The common characteristics were the physical deterioration of the patients, a rapid enlargement of lymph nodes, a decrease in the monoclonal IgM level, and a poor prognosis, with an exceedingly short median survival of 2 months.

Due to the rarity of HT in WM, the knowledge regarding this manifestation of clonal evolution has deepened only recently, with larger retrospective studies shedding more light on the unique features and associations [8–10]. The patients with transformed WM often present with extranodal involvement and high International Prognostic Index (IPI) scores. *MYD88* mutation status affects the incidence of HT and the prognosis of histologically transformed WM [10–12]. Despite the treatments that are similar to that used for de novo diffuse large B-cell lymphoma (DLBCL), the outcome of patients with HT remains dismal, even in the recent reports, with a median survival of 18 months following HT. The poor outcome appears to be primarily a consequence of the refractoriness of the transformed lymphoma or early relapses, including a high frequency of the central nervous system (CNS) involvement [11,13]. The usefulness of consolidation with an autologous stem cell transplantation (SCT) needs to be delineated and the role of novel agents that have dramatically impacted the outcome of patients with WM and other indolent B-cell lymphomas should be examined in the management of HT in WM.

## 2. Epidemiology and Risk Factors

Transformation to an aggressive B-cell lymphoma is estimated to occur in 1–4% of patients with WM [9,10]. The 5-, 10- and 15-year cumulative incidence rates of transformation were 1, 2 and 4% and 2, 5 and 6%, respectively, in the Dana-Farber Cancer Institute and the Mayo Clinic cohorts (Table 1) [9,10]. An increased incidence of HT among patients with WM treated with nucleoside analogs was suggested in retrospective studies [14], but this was not confirmed in the WM1 trial [15]. In the randomized WM1 trial that compared oral fludarabine and chlorambucil monotherapies as the frontline treatments, the 6-year cumulative incidence of HT was 8% in the fludarabine arm and 11% in the chlorambucil arm [15]. In the Mayo Clinic cohort, prior nucleoside-analog-based therapy or rituximab-based therapies did not impact the risk of transformation [10]. The rates of HT in patients treated with Bruton's tyrosine kinase (BTK inhibitors) have not been reported to date [16–19].

**Table 1.** Summary of the major published retrospective studies on HT in WM.

| Reference | 7 | 28 | 9 | 10 | 8 |
|---|---|---|---|---|---|
| Number of patients | 16 (including 14 from review of the literature) | 12 | 20 | 50 | 77 |
| Incidence of HT | NA | 13% | 2.4% at 10 years | 4.7% at 10 years | NA |
| No treatment prior to HT | 7% | 25% | 25% | 15% | 21% |
| Median time from MW to HT (years) | 4 | 3.7 | 4.4 | 4.5 | 4.6 |
| Male sex | 69% | 33% | 60% | 66% | 65% |
| Median age at HT | NA | 68 | 70 | 66 | 71 |
| Extranodal involvement | NA | 100% | 84% | 72% | 91% |
| Elevated LDH | NA | 80% | 67% | 53% | 72% |
| Front-line treatment for HT | | | | | |
| (R)-CHOP-like | NA | 33% | 80% | 80% | 85% |
| HyperCVAD | NA | 58% | 0% | 0% | 0% |
| Rituximab containing regimen | NA | 42% | 85% | 69% | 83% |
| Autologous SCT | NA | 8% | 30% | NA | 15% |
| Overall response rate | NA | NA | NA | 73% | 61% |
| Complete response | NA | NA | 77% | 53% | 48% |
| Progression-free survival (months) | NA | NA | NA | 10 | 9 |
| Survival after HT (months) | 2 | 75% died within 10 months | 32 | 38 | 16 |

CVAD: cyclophosphamide, vincristine, doxorubicin and dexamethasone; HT: histological transformation; LDH: lactate dehydrogenase; NA: not available; R-CHOP: rituximab, cyclophosphamide, doxorubicin, vincristine and prednisone; WM: Waldenström macroglobulinemia.

HT can occur at any time during the disease course, with a reported median time range to transformation of 4.3–4.6 years [8–11]. In retrospective studies, about 15–25% of patients were treatment-naïve at the time of the diagnosis of HT (Table 1) [8–10].

The data regarding the risk factors for the development of DLBCL in WM are scant. In recent years, two studies have demonstrated that the $MYD88^{WT}$ genotype was independently associated with a higher risk of HT [10,12]. HT occurred in 15% and 1% of $MYD88^{WT}$- and $MYD88$-mutated patients, respectively, in the study by Treon and colleagues, with 10-year cumulative incidence rates of 20% and 1%, respectively [12]. In the study by Zanwar and colleagues [10], $MYD88^{WT}$ was the only factor associated with an increased risk of HT in a multivariate analysis (odds ratio (OR) 7, *p* = 0.003). The $MYD88^{WT}$ genotype was also associated with a shorter time to transformation (hazard ratio (HR) 7.9, *p* = 0.001) (Figure 1). The patients with the $MYD88^{WT}$ genotype exhibit distinct patterns of somatic mutations affecting NF-kB signaling (*TBL1XR1*, *NFKBIB*, *NFKBIZ*, *NFKB2*, *MALT1*, *BCL10*, etc.), DNA damage repair (*TP53*, *ATM*, *TRRAP*) and epigenomic regulators (*KMT2D*, *KMT2C*, *KDM6A*). Many of these mutations occur in DLBCL, and might contribute to an increased risk of HT among patients harboring the $MYD88^{WT}$ signature [20].

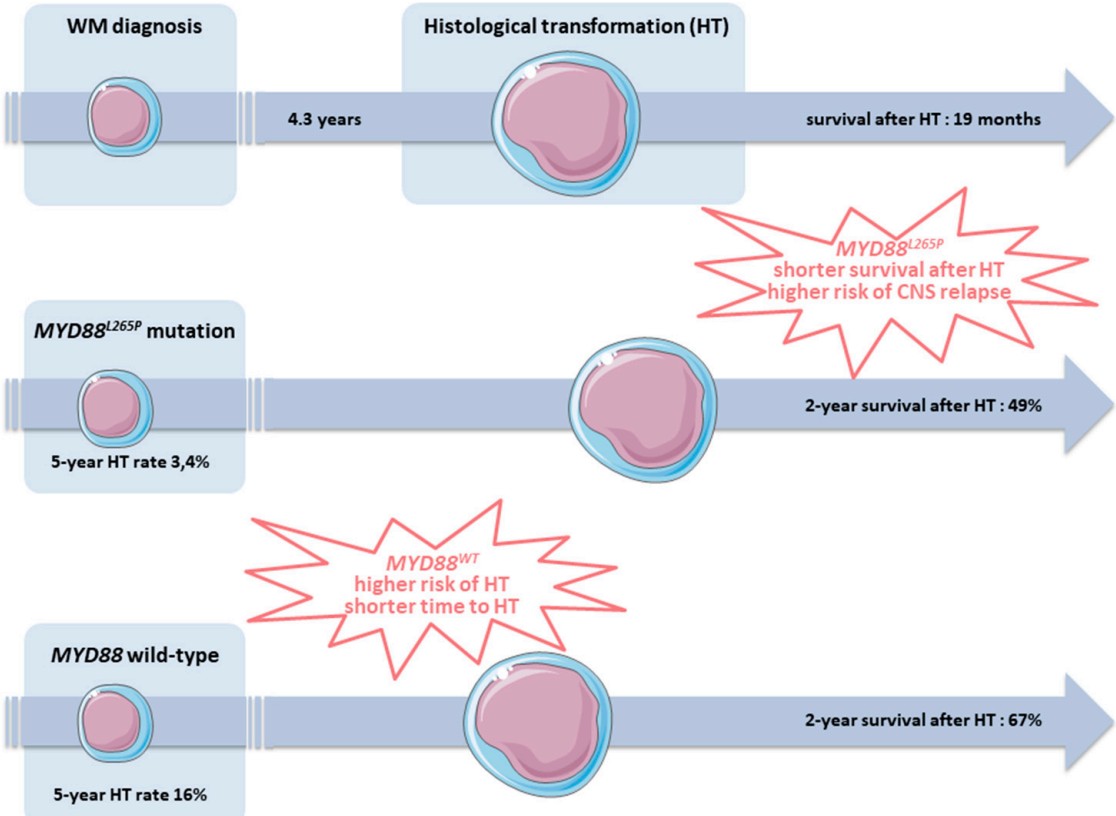

**Figure 1.** Schematic representation of HT in WM. Transformation rates, times to HT and survival rates after HT according to *MYD88* status.

### 3. Clinical Presentation and Diagnosis

HT should be suspected among patients with an established diagnosis of WM if they present with rapidly progressing constitutional symptoms, rising serum LDH levels or the appearance of extranodal disease. Most patients with HT present with an advanced Ann Arbor-stage disease and a high IPI score. Extranodal involvement is a common feature in transformed WM, being reported in 71 to 92% of patients (Table 1) [8–10]. This contrasts with WM, where extranodal and extramedullary involvement are rare (4.4%) [21]. The most common extranodal sites involved in HT include the bone and the bone marrow. Nonetheless, there is a relatively high frequency of CNS, testicular and skin involve-

ment [8,13,22]. The "de novo" DLBCL counterpart of the *MYD88*-associated sites (CNS, testis) has recently been recognized as a distinct entity, called "large B-cell lymphomas of immune-privileged sites" in the new World Health Organization (WHO) Classification of Hematolymphoid Tumors [23]. This category encompasses aggressive tumors of the activated B-cell (ABC) phenotype with concomitant *MYD88* and *CD79B* mutations and is associated with a poor prognosis [24]. Another frequently encountered observation in HT is a reduction in the serum IgM level [8,10], probably a reflection of the phenomenon of dedifferentiation (Table 2).

Similar to the approach for the other transformed indolent non-Hodgkin lymphomas, a tissue biopsy is a prerequisite to diagnose HT. The choice of the site of biopsy may be dictated by the $^{18}$fluorodeoxyglucose-positron emission tomography ($^{18}$FDG-PET/CT) results (Table 2) [25]. The transformed sites are expected to have a high maximum standardized uptake value (SUVmax), similar to that observed in DLBCL. In a French retrospective study, $^{18}$FDG-PET/CT-related data were available in 24 patients [8]. The median SUVmax was 15 (range, 4–38), and 71% presented with a SUVmax above 10. For comparison, in a study of 35 patients with non-transformed WM, 77% demonstrated $^{18}$FDG avidity on PET/CT with a mean SUVmax of 3 (range 1–8) [26]. Further studies are needed in WM to evaluate positive and negative predictive values of $^{18}$FDG-PET/CT for the diagnosis of HT.

## 4. Morphology and Clonal Evolution

HT in WM can be histologically heterogeneous. Cases of anaplastic large-cell lymphoma, T-cell lymphoma, plasma cell proliferation and EBV-associated DLBCL have already been described [27]. However, the largest retrospective series reported an overwhelming majority of DLBCL variants [8–10,28]. A few cases of aggressive lymphomas intermediate between DLBCL and Burkitt lymphoma or aggressive lymphomas that were not otherwise specified were described [8,10]. In the retrospective report by Lin and colleagues, the morphology of DLBCL is characterized by large B-cells with frequent mitotic figures, resembling centroblasts (75% of cases) or immunoblasts (25% of cases) [28]. Using the Hans algorithm [29], about 80–90% of cases are classified into the non-germinal center B-cell (non-GCB) subtype [8,10]. Based on immunohistochemistry, CD20 is positive in 95%, CD10 in 7–10%, BCL6 in 34–78%, MUM1 in 78–100%, BCL2 in 86–89% and MYC in 44–57% of patients [8–10]. The median Ki67 expression range is 80–90% [8,9]. EBV infection does not seem to be a pathogenetic trigger, since the majority of the transformed WM cases (83–100%) are negative for EBV-encoded RNA (EBER) in situ hybridization [8–10,28]. *MYC* gene rearrangement is detected in 11–38% of patients via fluorescence in situ hybridization (FISH) [8,10]. The prevalence of B-cell lymphomas with *MYC* and *BCL2* and/or *BCL6* rearrangements (referred to as "double-hit" or "triple-hit" lymphomas) in transformed WM is unknown, apart from a series of 7 cases where no rearrangement of *MYC*, *BCL2* or *BCL6* by FISH was found [30].

In chronic lymphocytic leukemia (CLL), most of the Richter transformations (80%) are clonally related to the CLL phase [31]. In transformed WM, an analysis of the rearrangement of IGHV-D-J genes is unavailable in the major retrospective studies. The light-chain expression was concordant between WM and transformed lymphoma in 75–100% of cases in these studies [8,10]. Using simultaneous *MYD88*$^{L265P}$ mutation and an IGHV analysis, a study on 4 cases of transformed and paired antecedent WM samples demonstrated that DLBCL can be clonally related to the WM or can occur as a new clone independent of WM (synchronous de novo DLBCL) [32]. More recently, 7 cases of transformed WM were analyzed using *MYD88*$^{L265P}$ mutation, IGHV rearrangement analysis and next-generation sequencing [30]. Three mechanisms of DLBCL development were proposed: (i) a "true" transformation with sequential mutation acquisition from WM to DLBCL; (ii) a clonal identity with a common origin but divergent evolution of WM and DLBCL; (iii) different lymphomas. A branching model of evolution has also been described with a transformed clone that did not evolve from the same subclone responsible for progression [33].

The data regarding the biology of the transformation of WM to DLBCL are limited. Jimenez and colleagues reported the results of whole-exome sequencing in 4 WM patients who transformed to DLBCL [33]. Their study revealed the genetic heterogeneity and complexity of HT. A much higher frequency of mutations is observed at the time of HT, inversely related to the time of transformation. They identify possible driver mutations present in a high proportion of tumor cells and conserved during transformation, as well as recurrent mutations gained at the time of HT (*PIM1*, *FRYL*, *PER3*, *PTPRD* and *HNF1B*). Another notable finding of this study is the potential role of *CD79B* mutations as biomarkers that could predict HT, given that *CD79B* has been found to be mutated in 3 of the 4 evaluated cases.

## 5. Prognosis

The outcome of patients with transformed WM is generally poor, with overall survival (OS) rates after the diagnosis of HT ranging from 16 to 38 months (Table 1) [8–11]. HT in WM is associated with an inferior OS compared to non-transformed patients. The Mayo Clinic series reported an increased risk of death, with an HR of 5.1 (95% CI 3.8–6.8, $p < 0.001$) [10], while the DFCI series found a median OS from diagnosis of WM to death of any cause of 9 years for patients who experienced HT versus 16 years for patients without HT [9].

In the French experience of 77 patients with transformed WM, the factors associated with shorter OS after HT were ≥2 lines of treatment for WM, prior rituximab exposure, time to transformation ≥ 5 years from the diagnosis of WM, elevated LDH and the absence of response to treatment (less than PR) in the univariate analyses [8]. In a multivariate analysis, the independent factors of OS were time to transformation ≥5 years and elevated LDH. A prolonged time to transformation is associated with previous exposure to therapy and possibly to immunological impairment related to WM itself. A similar prognostic impact of time to transformation was observed in Richter syndrome [34]. In transformed follicular lymphoma, the negative impact of previous therapy has also been described [35], but conflicting data exist on the prognostic value of the time to transformation [35,36]. Whereas a concurrent diagnosis of indolent lymphoma and DLBCL is usually associated with a better prognosis compared to subsequent development of HT [37], similar progression-free survival (PFS) rates and OS rates from the time of transformation were observed in transformed WM for patients with a synchronous versus sequential diagnosis of HT [8].

A recent international collaborative study developed and validated a prognostic index, called the tWIPI (Transformed Waldenström International Prognostic Index), with 2-year survival rate after HT as an endpoint [11]. Based on a training cohort of 133 patients evaluated between 1995 and 2016, three covariates were found to be independently predictive of 2-year survival after HT: elevated serum LDH (2 points), thrombocytopenia (platelet count $< 100 \times 10^9$/L; 1 point) and a history of any previous treatment administered for WM (1 point). Three risk groups were defined based on the total score: low-risk (0–1 point, 24% of patients), intermediate-risk (2–3 points, 59%) and high-risk (4 points, 17%), with 2-year survival rates of 81%, 47% and 21%, respectively (Table 2). This model was validated in an independent cohort of 67 patients and displayed high discrimination and calibration properties (Harrell C-index 0.75 in the training cohort and 0.79 in the validation cohort), unlike the IPI and the revised IPI (R-IPI) scores.

In Richter syndrome, the clonal relationship between the CLL and DLBCL clones is one of the most important prognostic factors: patients with clonally unrelated DLBCL (20% of cases) experience longer survival, similar to patients with de novo DLBCL [25]. The clonal relationship between WM and DLBCL was unknown in the large retrospective studies in this field, precluding any conclusion regarding its potential prognostic value in transformed WM [8–11].

The patients' *MYD88* mutation status at the time of the diagnosis of WM seems to have a prognostic impact on survival after HT (Figure 1) [11]. In the tWIPI study, 64 patients had available data for the *MYD88* mutation status. Patients with *MYD88*$^{L265P}$ mutation

had a significantly lower 2-year survival rate after HT compared to those with $MYD88^{WT}$ disease (49% vs. 67%, *p* = 0.018) (Table 2). This finding, in line with previous studies in de novo DLBCL [38,39], should be confirmed in larger cohorts of patients. The presence of $MYD88^{L265P}$ mutation has also been associated with a higher rate of CNS relapse in transformed WM (17% versus 0% for $MYD88^{WT}$ patients) (Figure 1) [13]. In this study, the median survival after CNS relapse was 6 months.

**Table 2.** Key points on clinical presentation, diagnosis and prognosis of HT in WM.

| | |
|---|---|
| **Clinical presentation** | ✓ High frequency of extranodal involvement, in particular skeletal bone, bone marrow and *MYD88*-associated immune-privileged sites (CNS, testis, skin) <br> ✓ Advanced stage and high IPI score <br> ✓ Elevated serum LDH <br> ✓ Decrease in serum IgM level |
| **Diagnosis** | ✓ Suspicion of HT in case of physical deterioration in patients with WM, rapid growth of lymph nodes, extranodal involvement or rise in LDH level <br> ✓ Tissue biopsy required for diagnosis of HT <br> ✓ Tissue biopsy may be directed by 18FDG-PET/CT |
| **Prognosis** | ✓ Poor outcome after HT <br> ✓ Prognosis index (tWIPI) based on 3 predictors of 2-year survival after HT: elevated LDH (2 points), platelet count $< 100 \times 10^9$/L (1 point) and any previous treatment for WM (1 point) <br> ✓ Presence of $MYD88^{L265P}$ mutation: lower 2-year survival after HT and higher risk of CNS relapse |

CNS: central nervous system; $^{18}$FDG-PET/CT: $^{18}$fluorodeoxyglucose-positron emission tomography/computed tomography; HT: histological transformation; IPI: International Prognostic Index; LDH: lactate dehydrogenase; tWIPI: Transformed Waldenström International Prognostic Index; WM: Waldenström macroglobulinemia.

## 6. Treatment Options

Given the disease's rarity, prospective trials have not been conducted in this setting. Moreover, the patients with transformed WM are usually excluded from clinical trials or represent a minority of patients among transformed indolent lymphomas. Accordingly, no recommendations exist in the literature on the treatment of transformed WM.

### 6.1. Chemo-Immunotherapy

The treatment of transformed WM usually mirrors that of DLBCL, but with inferior outcomes. Data on the response rates and outcomes are based on retrospective studies. The most commonly reported initial regimen used to treat patients with HT is R-CHOP (rituximab, cyclophosphamide, doxorubicin, vincristine, prednisone)-like chemo-immunotherapy (CIT, 62% to 85%) (Table 3) [8–10]. The overall response rates following R-CHOP-like regimen range from 61% to 79%, and the complete response (CR) rates range from 48% to 77% [8–10]. The response duration is, however, short, with median PFS rates of 7 to 10 months (Table 1) [8,10]. Data on more aggressive CIT regimens, such as R-EPOCH (rituximab, etoposide, prednisone, vincristine, cyclophosphamide and doxorubicin) or ACVBP (adriamycine, cyclophosphamide, vinblastine, bleomycin, prednisone), are too sparse to draw any meaningful conclusions. In the study by Lin and colleagues, 7 of 12 patients were treated with hyper-CVAD (fractioned cyclophosphamide, vincristine, doxorubicin and dexamethasone); 6 died within the first 5 months, while 1 was alive at 8 months after consolidation via BEAM treatment (carmustine, etoposide, cytarabine and melphalan) and autologous stem cell transplantation (autoSCT) [28]. Other primary therapies that have been used include DHAP (dexamethasone, cytarabine and cisplatin), ICE (ifosfamide, cyclophosphamide and etoposide) and GEMOX (gemzar and oxaliplatin), in particular for patients who were previously treated with CHOP +/− R for WM, patients with CNS involvement or patients presenting a contraindication to anthracyclines.

About 5% of the patients were managed with palliative care, reflecting the underlying comorbidities and frailty of this population.

### 6.2. Central Nervous System Prophylaxis

Relapse within the CNS occurs in 2% to 5% in DLBCL and is associated with a poor prognosis with a median OS of 5–6 months [40–42]. The incidence of CNS relapse in transformed WM has been recently reported, with a 3-year rate of 11%, similar to that observed in the CNS-IPI high-risk group [13,40]. The optimal prophylactic strategy is unknown. There is growing evidence suggesting that intrathecal therapy is ineffective [43] and recent studies challenge the notion that high-dose methotrexate (HD-MTX) may provide a benefit [44,45]. However, CNS prophylaxis is a major concern in transformed WM, particularly in patients with kidney or adrenal involvement or an $MYD88^{L265P}$ mutation, for whom an even higher incidence of CNS relapse has been observed. If HD-MTX is considered, recent studies in DLBCL suggest its delivery could be deferred beyond cycle 1 of R-CHOP (on day 1 and especially before day 10) or even until R-CHOP completion to avoid toxicities or R-CHOP delays [46,47].

### 6.3. Stem Cell Transplantation

High-dose chemotherapy (HDC) followed by autoSCT in the first CR (CR1) has been proposed as a consolidation approach in fit patients with transformed indolent lymphomas, based on the understanding that HT has a poor prognosis, but its use is still debated, as the evidence regarding the effectiveness of such an approach is weak [48–53]. The studies dealing with this modality are usually retrospective with heterogeneous populations (autoSCT in CR1 or later, treated versus untreated indolent lymphomas, previous use of rituximab versus not, etc.) and mainly include transformed follicular lymphomas [49–53]. Transformed WM represents less than 5–10 patients in these reports; therefore, there are no consistent data on the response, PFS and OS rates. For example, only 2 patients with transformed WM were treated with allogeneic SCT (alloSCT) in a study of 34 patients with transformed non-follicular indolent lymphomas treated with auto or alloSCT [48] and only 1 patient in a recent study of 49 patients exploring the role of autoSCT in first remission [54].

In the study by Lin and colleagues, only one patient underwent HDC with BEAM (carmustine, etoposide, cytarabine and melphalan) treatment and autoSCT after hyperC-VAD [28]. In the Dana-Farber cohort, 6 of 20 patients received autoSCT, without a survival benefit ($p = 0.13$) [9]. Nevertheless, 5 patients underwent autoSCT at the time of HT relapse and only 1 in CR1. The Mayo Clinic study also found no positive impact on survival for patients managed with autoSCT ($p = 0.4$), but again a small proportion of patients (3 out of 50) received it as a consolidation approach after the initial therapy [10]. In a series of 77 patients with transformed WM, 10 (13%) underwent autoSCT, including 7 after first-line treatment for HT [8]. When comparing the responders to frontline therapy, a plateau seems to emerge for patients receiving autoSCT, with the median OS not being reached (versus 4.5 years for the responders to the on-transplant-based approach), although statistical significance was not attained ($p = 0.33$), probably due to the small number of patients.

Whether consolidative autoSCT at the time of frontline therapy for transformed WM should be recommended for the eligible patients remains to be elucidated. The optimal way to analyze the role of autoSCT in transformed WM would be a randomized trial, but such an approach is not practical and unlikely to be undertaken, given the rarity of this condition. In the absence of a randomized controlled trial, the effectiveness of autoSCT may be evaluated by comparing the cohort treated with autoSCT during first remission with a matched cohort of patients with chemosensitive disease that did not undergo autoSCT. Nevertheless, most patients with transformed WM are unfit (median age at the time of HT of 69 years with comorbidities) or do not achieve an adequate response to proceed to autoSCT. The role of alloSCT in transformed WM is unclear.

**Table 3.** Key points on the treatment of HT in WM.

| |
|---|
| ✓ Treatment with similar regimens used in de novo DLBCL (R-CHOP-like regimen) |
|     ○ ORR 61–79% |
|     ○ CR 48–77% |
|     ○ PFS 7–10 months |
| ✓ No sufficient data on more aggressive chemo-immunotherapy regimens |
| ✓ CNS prophylaxis should be considered (HD-MTX) |
| ✓ Autologous SCT as consolidation in fit patients responding to induction chemotherapy should be considered |
| ✓ No sufficient data on allogeneic SCT, novel agents, CAR T-cells |

CAR: chimeric antigenic receptor; CNS: central nervous system; CR: complete response; DLBCL: diffuse large B-cell lymphoma; HD-MTX: high-dose methotrexate; HT: histological transformation; ORR: overall response rate; PFS: progression-free survival; R-CHOP: rituximab, cyclophosphamide, doxorubicin, vincristine and prednisone; SCT: stem cell transplantation; WM: Waldenström macroglobulinemia.

*6.4. Novel Agents*

Given the efficacy of BTK inhibitors, in particular ibrutinib and zanubrutinib [16–19], and BCL2 inhibitors such as venetoclax [55] in WM, these agents could represent potential therapeutic options in transformed WM. Ibrutinib induces durable responses and long-term disease control in WM [17,18]. Its efficacy has been demonstrated both in treatment-naïve and previously treated patients with WM, with overall response rates of 100% and 90.5%, respectively, and median PFS not reached [17,18]. The phase 3 ASPEN study has shown that zanubrutinib was at least as efficient as ibrutinib and less toxic (less cardiovascular toxicity) [16]. BCL2 overexpression and $MYD88^{L265P}$ mutations are found in up to 90% and 67% of transformed cases, respectively. The combination of ibrutinib with R-CHOP was evaluated in a phase 3 study in ABC DLBCL [56]. The event-free survival (EFS), PFS and OS rates were only improved in patients younger than 60 years. Increased toxicity was observed among the patients older than 60 years. The efficacy and safety of venetoclax associated with R-CHOP was assessed in the phase 2 CAVALLI study; the combination demonstrated increased myelosuppression but potentially improved outcomes in BCL2+ subgroups [57].

*6.5. CAR T-Cells*

CD19-targeted chimeric antigen receptor (CAR) T-cell therapies have demonstrated substantial efficacy and can lead to durable responses in relapsed/refractory (R/R) DLBCL, including transformed follicular lymphomas [58,59]. These therapies were approved for third-line R/R DLBCL and their role as second-line treatments has been recently published [60,61]. In R/R WM, the activity of CD19-directed CAR T-cell therapy has been recently reported with preclinical and clinical data in 3 heavily pretreated patients [62]. However, patients with transformed WM were not included in these studies. The study by Abramson and colleagues on lisocabtagene maraleucel included patients with DLBCL transformed from any indolent lymphoma. Eighteen patients had non-follicular transformed indolent lymphomas, of which only 2 were transformed WM, with no patient-level data available [63]. The potential effectiveness of the CD19-directed CAR T-cell therapy in R/R-transformed WM has recently been suggested in a case report of a 71-year-old man who had received 2 prior lines of therapy for WM and was then treated with R-CHOP in combination with ibrutinib at the time of HT (high-grade B-cell lymphoma with *MYC* and *BCL6* rearrangements). He relapsed after 18 months and received R-DHAP and BEAM with autoSCT. His disease was progressive 1 month after autoSCT and he was treated with axicabtagene ciloleucel following fludarabine and cyclophosphamide. CR (PET/CT and bone marrow biopsy) was achieved and maintained at 1 year [64]. A longer term follow-up is needed and studies on more patients with transformed WM would be informative.

## 7. Proposed Management of HT in WM

The recommendations for the optimal management of HT in WM are not easy to formulate, given that the available information is primarily derived from retrospective studies and case reports. We propose the following approach. Clinicians should remain highly vigilant for this complication and should further evaluate patients with WM who develop physical deterioration, the setting of rapidly enlarging lymph nodes, extranodal involvement, a sudden increase in serum LDH levels or a paradoxical decrease in the serum IgM spike. A tissue biopsy is mandatory to diagnose HT and may be dictated by the findings of an $^{18}$FDG-PET/CT scan, which is recommended for all patients suspected to have HT. If possible, participation in a clinical trial should be considered for treatment. Otherwise, a CIT such as R-CHOP may be recommended as the primary therapy. CNS prophylaxis with HD-MTX should be considered if feasible and consolidation with autologous SCT should be discussed in fit patients who respond to CIT.

## 8. Conclusions and Perspectives

Recent retrospective studies have deepened the understanding of transformed WM. However, the outcome of patients is still poor, and due to the rarity of this complication, HT in WM is an area of unmet need. Advances in understanding the biology of transformed WM are needed in order to expand the currently available suboptimal therapeutic options and to improve the outcome. The wider use of novel agents such as BTK inhibitors in WM might change the epidemiology of HT and will require more studies. We should also be aware of the frailty of these patients with HT, with potential for experiencing increased toxicity with CIT and a consolidative strategy. Even more than in other areas, international collaborative efforts should be pursued to improve our knowledge of this rare condition.

**Author Contributions:** Contribution: E.D. and A.D. wrote the manuscript. Final approval of manuscript: E.D., C.T., E.T., P.M., D.T., P.K., J.J.C. and A.D. All authors have read and agreed to the published version of the manuscript.

**Funding:** This research received no external funding.

**Institutional Review Board Statement:** Not applicable.

**Informed Consent Statement:** Not applicable.

**Data Availability Statement:** Not applicable.

**Conflicts of Interest:** P.M. received research funds and consulting fees from Beigene, Janssen. D.T. received research fees and honoraria from Janssen, Roche, Beigene, Takeda, EUSA and CSL. P.K. is a PI of studies for which the Mayo Clinic has received research funding from AbbVie, Sanofi, Amgen, GSK, Ichnos, Takeda, Regeneron and Karyopharm. P.K. has received honoraria from X4 pharmaceuticals, Beigene, Pharmacyclics, Imidex, Clinical Care Options, GSK, Oncopeptides, Cellectar and Karyopharm. J.J.C. received research funds and consulting fees from Abbvie, AstraZeneca, Beigene, Janssen, Pharmacyclics, Polyneuron, Roche and TG Therapeutics.

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
