# Peer review of "Transformed Waldenström Macroglobulinemia: Update on Diagnosis, Prognosis and Treatment"

_hemato, doi:10.3390/hemato3040044_

Round 1

Reviewer 1 Report

For WM, treatment with  BTKis is getting more important therapy. But the authors mentioned a little in the section 6.4. So, they should expand this issue also in this review paper.

Author Response

We thank the reviewer for its useful comments that we have taken into account. We agree with Reviewer 1 on the importance of BTKis in WM. We expanded data on efficacy of BTKis in the section 6.4.

Reviewer 2 Report

The present manuscript describes characteristics, treatment and outcomes of patients with histological transformation of WM.

It is well written and gives a comprehensive overview. There is only one minor point: The author often refer to Richter transformation. The respective paragraphs are in part difficult to read and do not add substantial additional information. It can be considered to delete these sentences.

Author Response

We thank the reviewer for its comments and suggestions. We only refer to Richter transformation in sections 4 (morphology and clonal evolution) and 5 (prognosis) to introduce the clonal relationship between indolent lymphoma and aggressive transformation. We do think it is important to draw a parallel with Richter transformation because similarities exist between transformed WM and Richter in terms of survival and prognostic factors. Clonal relationship is unknown in the large retrospective studies on transformed WM but some case reports have shown that DLBCL can be clonally related to WM or occurs as a new clone. It probably has prognostic importance but data on transformed WM are too scarce to make recommendations on how to treat patients according to clonal relationship. Moreover, in an early version of the manuscript, we made more references to data on other transformed indolent lymphomas (CLL, follicular lymphoma) but the whole manuscript was more difficult to read. We decided to keep only the sentences on Richter syndrome and clonal relationship.